# Innovative Human Three-Dimensional Tissue-Engineered Models as an Alternative to Animal Testing

**DOI:** 10.3390/bioengineering7030115

**Published:** 2020-09-17

**Authors:** Patrick Bédard, Sara Gauvin, Karel Ferland, Christophe Caneparo, Ève Pellerin, Stéphane Chabaud, Stéphane Bolduc

**Affiliations:** 1Faculté de Médecine, Sciences Biomédicales, Université Laval, Québec, QC G1V 0A6, Canada; patrick.bedard.2@ulaval.ca (P.B.); sara.gauvin.1@ulaval.ca (S.G.); karel.ferland.1@ulaval.ca (K.F.); 2Centre de Recherche en Organogénèse Expérimentale de l’Université Laval/LOEX, Centre de Recherche du CHU de Québec-Université Laval, Axe Médecine Régénératrice, Québec, QC G1J 1Z4, Canada; christophe.caneparo.1@ulaval.ca (C.C.); eve.pellerin.1@ulaval.ca (È.P.); stephane.chabaud@crchudequebec.ulaval.ca (S.C.); 3Département de Chirurgie, Faculté de Médecine, Université Laval, Québec, QC G1V 0A6, Canada

**Keywords:** tissue engineering, scaffold, extracellular matrix, epithelium

## Abstract

Animal testing has long been used in science to study complex biological phenomena that cannot be investigated using two-dimensional cell cultures in plastic dishes. With time, it appeared that more differences could exist between animal models and even more when translated to human patients. Innovative models became essential to develop more accurate knowledge. Tissue engineering provides some of those models, but it mostly relies on the use of prefabricated scaffolds on which cells are seeded. The self-assembly protocol has recently produced organ-specific human-derived three-dimensional models without the need for exogenous material. This strategy will help to achieve the 3R principles.

## 1. Introduction

### 1.1. A Brief History of Animals as Models in Science

Scientists have used animal models in research at least since the Ancient Greek Era. For example, Alcmaeon of Croton studied the brain with dogs in the 6th century B.C. [1]. Since then, many different models have been used for different applications: the nematode *Caenorhabditis elegans*, the fly *Drosophila melanogaster*, zebrafish, frog *Xenopus*, mouse, rat, rabbit, dog, and primate rhesus macaque are some examples of models that have been used and which are still used today in research [2,3,4,5,6] (Figure 1).

The mouse and the rat are, by far, the most used animals (Figure 2). Animal models have improved in the past few years, and these modified animals now present more accurate models to medical research. For example, scientists can modulate animal genomes by adding or deleting genes to mimic disease or to study the function of these genes [12]. These animals are called transgenic animals. The most common transgenic animal is the transgenic mouse, but this technology applies to other species like rats, cats, and rabbits [1,13,14,15,16].

### 1.2. Why Animal Testing

There are many reasons why scientists all over the world still use animals to study human diseases. First, non-human animals share genetic and physiologic similarity to humans. For example, mice share 80% of their genome with humans [17]. Since animals are very similar to humans, they can suffer diseases similar to human ones. For example, Joseph von Mering and Oskar Minowski have used a dog pancreas to prove that the pancreas has a role in diabetes [18]. Another advantage of an animal models is that while they often have a shorter life expectancy than humans, they generally share a similar ageing pattern; therefore, scientists can study disease in a lifetime in less time than in a human body [13]. Moreover, some pathologies or biological phenomena, requiring interaction between several organs, do require the use of animals as an experimental “unit”: for example, studying metastases. In addition, despite the intense work of lobbying from opponents to animal testing, many regulatory agencies still require the use of animals in preclinical testing phases. For example, U.S. federal laws require that non-human animal research occur to show the safety and efficacy of new treatments before any human research will be conducted (U.S. Food and Drug Administration. Investigational New Drug (IND) Application. U.S. Food and Drug Administration. 5 October 2017. Available at the US Food and Drug Administration website.

### 1.3. Relevance of 3R Principles in Research

If animal models can be seen as valuable tools to study human diseases, they nevertheless have several flaws. For example, more ethical concerns have been raised about the use of living things in research. In 1959, Russel and Burch defined the 3R principles for animal use in research: replacement, reduction, and refinement. To offer better treatment to laboratory animals, 3Rs were proposed in “the principle of human experiment technique” [19]. Replacing should be the primary goal of the 3Rs: if the research no longer uses animals, the problem related to their use disappears. Nevertheless, as this seems unrealistic in the short or medium term, reducing and refining should be short/middle-term goals. The term “reduce” means minimization of the number of animals that are used in research and the term “refine” corresponds to the use of techniques that are doing less harm to animals [20].

### 1.4. Why Replacing, Reducing, and Refining?

#### 1.4.1. Replacement

In several cases, animal models should be replaced by more accurate and innovative in vitro models. First, there are many ethical concerns about animal use in research [21].

Moreover, the translation of results obtained from animal models to humans has long been questioned. In some cases, results observed in animals do not accurately predict human responses, like it has been the case for different toxicity tests that did not succeed in accurately predicting toxicity in humans [22]. The complexity of whole mammal organisms is also an aspect to consider [23]. Although animal models have been improved over time, for instance, to study specific conditions, there are still uncontrollable variables in animal organisms [24]. The reproducibility of research involving animal models has also been considered inferior, especially in preclinical research [25]. Because of the possible lack of relevant animal testing results, this practice has been qualified wasteful by some [26,27]. For example, the chimpanzee was considered an excellent model to develop a human immunodeficiency virus (HIV) vaccine in the 1990s. Some vaccines were promising in the primate, but the results were not the same in human trials. Indeed, the chimpanzee does not have the same immune system as humans. Still today, no vaccine has been found to prevent HIV [28]. The mouse model for cancer study is another example of the lack of accuracy of animal models. Many studies have described an elegant cure to treat cancer in mice, but very few of these results are translatable to humans [29].

Economic considerations also encourage the need for a change toward alternative in vitro models. The cost of animal breeding/housing for scientific purposes reveals high expenses and is an essential factor to consider [30,31]. In addition, animal protocols can be time-consuming and require skilled and trained operators to perform specific experiments [32].

The replacement of animal testing in research can be achieved through various alternatives [31]. These can sometimes bring entirely new information that cannot be obtained from studying animals, hence the importance of improving animal testing alternatives.

#### 1.4.2. Reduction

In 2018, 780,070 animals were used for animal research in the United States [10]. This number excludes many species like mice and rats, which are not protected by the animal welfare act. The animal welfare act does not protect ninety-three percent of animals used in research; the estimation of actual animal use in research in 2018 is between 11 and 23 million. This is a slight increase compared to 2017; nevertheless, the number of animals used in research has been reduced by more than half since 1985 [10].

Various reasons led the industry to reduce animal testing: not only the ethical concerns related to the pain and distress animals can experience during protocols [33] but also the possible drawbacks of these models. Once again, saving experimental animal numbers is saving money.

The need to reduce animal testing has also arisen through legislations requiring adherence to the 3Rs. Since 2006, the EU Registration, Authorization, and Restriction of Chemicals (REACH) legislation has addressed animal testing. It aims to reduce the number of animal tests to a minimum while promoting alternatives [34]. Currently, several in vitro protocols, including tissue engineering-derived ones, have been designed to reduce animal use, especially for toxicological testing. For example, protocols using 3D reconstructed human cornea-like epithelium and epidermis models were set up as an alternative to in vivo Draize rabbit eye and skin irritation tests [35]. Several alternatives will be described in Section 1.5.

#### 1.4.3. Refinement

Refinement is a crucial topic in research using animal models to improve animal welfare. Animals can feel pain and mental distress. For example, in 2010, a research team correlated the level of pain that mice were experiencing with their facial expressions in order to provide quantifiable data [36]. Providing a better quality of life to animals is especially important because animals’ discomfort and distress during experiments can lead to fluctuations in the results and require repetition of these experiences, then increasing the number of animals. Briefly, the objective is, not exhaustively, to optimize the methodology applied to animals; this involves reducing, eliminating, or relieving their pain or distress, and thereby improving their wellbeing: to improve transport, breeding, and housing conditions, to plan the protocol to avoid stress, to use painless or non-invasive procedures, to provide adequate care before, during, and after the operation, to balance anesthesia procedures with regard to pain relief, to minimize the duration of certain studies to the strictly required period, and to choose more appropriate euthanasia procedures. The analyses should be carefully planned to obtain the maximum amount of data from the experiments.

However, this could not be directly modified by the use of tissue engineering (TE), which is the present subject. Many good reviews treat this subject [37,38].

### 1.5. Alternatives to Animal Models

#### 1.5.1. Mammalian Cell Culture

Cell culture is a promising approach to complement animal data or completely substitute it [31]. In these systems, cells originating from animal or human biopsies are grown in an artificial environment and are subjected to controlled conditions, including parameters such as pH, osmolarity, temperature, humidity, and gaseous atmosphere. A culture medium is used to provide cells with nutrients (amino acids, vitamins, inorganic salts, and carbohydrates). However, it is generally further supplemented with proteins obtained from animal-derived serum as a source of growth factors and hormones [39]. Cell culture represents an indispensable tool to improve our understanding of cell biology and in vivo cell behaviour mechanisms. It is making rapid progress and its development has the potential to lead to a decrease in animal use [31]. Different cell culture techniques, including 2D and 3D methods, have been developed (see below). This follows a trend towards the development of models that better mimic the cellular microenvironment.

#### 1.5.2. Plant Tissue-Based Materials

Toxicity assays in plants can be used for preliminary in vitro studies as a substitute for animals. While it is more complex to relate plant-based data to human research, different assays have been useful for safety assessment. For example, the Allium genotoxicity test has been useful for the screening of pesticides, water pollutants, or industrial waste chemicals [40]. Likewise, tests using pollen suspensions (pollen tube growth tests) have been reliable in some instances. For example, this test has been used to assess the safety of different mouthwashes [41]. However, plant substitutions’ applications and success stay limited [42].

#### 1.5.3. Yeasts-Based Assays

Yeast *Saccharomyces cerevisiae* has been used for the screening of potential genotoxic substances for over 40 years, thus contributing to reducing the large number of animals used in genotoxicity tests. Chemicals from anticancer drugs and pharmaceuticals, cosmetics, and insecticide residues are quickly and efficiently screened to detect any mutagens. Despite their limited capabilities, tests held on this unicellular eukaryote are being upgraded continuously, for example, by increasing the permeability of yeast cell walls [43].

#### 1.5.4. In Silico Methods

Databases, computational models, or simulations are being actively developed to improve advances in medicine. Software can be used to help make decisions, predictions, and hypotheses. This way, it is possible to replace or reduce the use of in vivo tests by minimizing the need to test directly on animals. In pharmacology, in silico methods have been highly used for toxicology tests and drug screening. This allows high throughput screening and reduces time and cost in drug discovery [44]. With their rapid development, in silico technologies are considered to have a bright future [45].

#### 1.5.5. Other Methods

Other alternatives, such as reconstructed enzyme systems [31], microbiological systems (from bacteria or fungi), DNA chips, or microfluidic chips [42] are being used, but they will not be further discussed in this text, as we focus on providing an overview of cell culture techniques and the relevance of their development as an alternative to animal testing in agreement with the 3R principles.

### 1.6. Cell Culture

#### 1.6.1. Two-Dimensional (2D) Cell Cultures

Traditional 2D cell cultures have been used as in vitro models for over a century [46]. In these systems, cells are grown as a pure monolayer directly on plastic or glass flat dishes. The cells are in contact with a nutrient fluidic medium that can be modified to study various parameters. The simplicity and efficiency of 2D platforms have contributed to their general acceptance by the scientific community, and they are now a well-established practice [47]. For instance, they offer a relatively easy method to investigate cell behaviour via imaging or gene expression profiling. Their efficiency allows for high throughput screening (HTS) in drug discovery [48], which offers the potential to reduce animal use [31]. With these advantages, 2D models seem to represent an exciting method to replace animal testing. However, the limitations of 2D platforms have been highlighted throughout the years. The main limitation of these models is that they do not accurately mimic the natural 3D organization of cells and their extracellular matrix (ECM), thus varying the cells’ microenvironment from the one found in vivo [49,50,51]. Consequently, cell–cell and cell–ECM interactions are not as representative as in vivo. Since cell behaviour, like cell differentiation, migration, morphogenesis, and proliferation, has been proven to be highly influenced by its biochemical and biomechanical microenvironments [47], 2D cell culture can lead to results that deviate from in vivo responses [52,53].

Various factors are responsible for the lack of accuracy between 2D and in vivo microenvironments.

The homogenous concentration of nutrients: As monolayers, cells obtain similar access to nutrients and growth factors from the medium. However, in vivo, the concentration of soluble factors that affect cell differentiation, migration, and communication is distributed through dynamic spatial gradients [54]. While this characteristic can make 2D cell culture an advantage for some research works, it does not allow for an accurate representation of in vivo conditions.

Substrates’ rigidity: In 2D models, cells are grown directly on plastic or glass supports, which are rigid and stiff platforms. This does not allow for natural growth kinetics and cell attachments [49].

Planar environment: Growing cells directly on flat surfaces is unnatural and affects cell behaviour in various ways. Firstly, it promotes an artificial polarity between the apical and basal surfaces of cells [49]. This is mainly a problem when growing physiologically non-polar cells. A planar environment also restricts the development of complex morphologies that can usually deploy in a 3D environment. The migration of cells is also impacted because the lack of ECM causes a decrease in the resistance to migration.

Use of animal serum: Even if it is not specific of 2D cell cultures (it is also found in several 3D cell culture models), the use of animal serum, especially fetal serum, can induce cell reactions, which differ from the in vivo physiology. For example, it can mask several signaling pathways by activating other pathways. Moreover, animal serum contains not only cytokines and hormones but also the remainder of the extracellular matrix, bioactive lipids, etc., depending on how the serum had been prepared: the composition of animal serum can differ from lot to lot and provide inconsistent results. All these aspects will be discussed later in the text.

These deviations of the 2D microenvironment from the in vivo one lead to various problems. Misleading and non-predictive data for in vivo responses coming from 2D models are causing failure of clinical trials, which are expensive phases of drug development [53]. In addition, some essential characteristics of cancer cells cannot be appropriately modeled in 2D platforms because of their lack of efficiency to mimic physiological cell behaviours like migration [55]. Therefore, to accurately study various cell behaviours in vitro, 2D systems should be improved to models that replicate 3D in vivo organization to achieve a better imitation of the biochemical and biomechanical properties of the ECM. The more accurate the model is to the natural microenvironment, the more it could be relevant to replace animal testing.

The search for new culture platforms to improve physiologically relevant microenvironments in cell culture to solve these problems led to the development of so-called 2.5D systems that are considered midway between 2D and 3D cultures. There are different opinions on what consists of a 2.5D system [56,57]. However, Zerda et al. and Shamir et al. define 2.5D culture as growing cells on either biological or synthetic coating instead of directly on the plastic. Materials used as a coating represent the ECM and usually have topographical features, unlike plastic. This impacts the architecture of cells by modifying the cell membrane curvature, especially on the basal surface. A more physiologically relevant architecture of the tissue is developed that way [51]. Therefore, 2.5D models differ from 2D models by inducing curvature on basal cell surfaces, whereas basal surfaces of cells grown on 2D plastic surfaces are usually flat. However, it cannot be considered a 3D system as cells are not entirely embedded in the ECM and bear fluid facing surfaces, while this would not be the case in vivo. This results in an apical and basal polarity of cells and the modification of cell interactions by diluting cell-produced factors in the fluid [51], just like for 2D. While 2.5D models better replicate some features found in vivo than 2D classical models, they still face limitations in their representation of the natural microenvironment.

#### 1.6.2. Three-Dimensional (3D) Cell Cultures

Three-dimensional cell culture represents an excellent alternative to the animal model. A 3D cell culture is defined by a cell culture that can mimic a living organ’s organization and microarchitecture [58]. Many methods have been developed in the last two decades to obtain 3D cell cultures (Figure 3). These methods will be presented further. The 3D cell culture market is expanding every year; it is projected to reach 1846 million USD in 2024 [58]. Every year, techniques improve; scientists can now make skin, cornea, blood vessels, and many more with 3D cell culture [59,60,61].

#### 1.6.3. Advantages of 3D Cell Culture Compared to 2D

The improvement of classical 2D cell culture to 3D systems offers an increasing opportunity to replace animal models. While 2D models could lead to misleading results and non-predictive data for in vivo responses because of their lack of accuracy to mimic an in vivo microenvironment [53], 3D cell culture succeeds to replicate a higher number of in vivo features [50].

With its three-dimensional organization, this method overcomes various disadvantages encountered in 2D-cultured cell microenvironments. Factors improved by a three-dimensional environment, where cells are surrounded by ECM, include heterogeneous access to soluble factors, nutrients, and oxygen [54,62], a more in vivo-like generation of apical–basal polarity [49], and better replication of mechanical stress by growing in a less stiff environment [49]. This allows for more relevant cell–cell and cell–ECM interactions [50] and contributes to making 3D-cultured cell behaviour more reflective of in vivo cellular responses. Overall, it has been found that these systems can promote better replication of cell morphology [63] and adhesion [49]. They can also enhance cell differentiation, proliferation, and migration [63] in addition to impacting cell survival [64] and gene expression [52]. This might enhance tissue-specific functions in 3D platforms.

Three-dimensional systems can be used to mimic more accurately specific disease models [65,66,67,68]. For instance, cancer models cannot be modeled with 2D classical cell culture as this method fails to accurately replicate essential characteristics of the tumour microenvironment like its dynamic and complexity. In a study by Karlsson and collaborators, 3D systems showed better resistance to anticancer drugs than in monolayer cultured cells, showing that results are profoundly impacted by a three-plane environment [69].

Therefore, although 2D models have contributed to highly increase our general understanding of cell behaviour, 3D cell culture can provide platforms that better replicate the in vivo environment for more reliable and predictable results for human application. Three-dimensional platforms are suitable for a wide range of applications, and numerous 3D cell culture techniques are being actively developed. This point goes in agreement with the 3R principles, which aim to replace and reduce animal use.

#### 1.6.4. Advantages of Cell Culture Compared to Animal Models

Besides sparing animal use and leading the way for a more human approach for testing, in vitro cell cultures have additional advantages that are not met when using animal models. Cell culture allows for better control of variables, whereas in vivo, many variables are uncontrolled, and their effects are sometimes unknown due to the complexity of organisms [23]. Thus, cell culture can enhance reproducibility and facilitate the study of cellular and molecular mechanisms [70,71].

An essential advantage of cell culture as an alternative to animal models is that human-based models can be grown by using human cells, thus possibly minimizing the questionable translation of in vitro results to humans. Furthermore, this opens the door for personalized medicine development. Autologous patients’ cells can be grown for drug testing [72], disease modeling [65,73,74,75], and could even be useful to assess diagnoses [66]. Tissue engineering can also provide autologous tissues for patients in need of grafts for surgical reconstruction. Because of high risk of immunologic rejection and public health risk concerns, animal tissue transplant in humans is greatly limited [76,77]. Cell culture overcomes many graft limitations, like the lack of healthy tissue from a donor site, by allowing the growth of autologous and physiologically relevant tissues for grafts [78,79].

However, cell culture cannot wholly substitute animal use. Whole organisms are necessary to explore interactions between organs and physiological functions. This explains the recent emergence and improvement of 3D cell culture techniques to achieve the development of more complete and accurate in vitro systems following the objective of minimizing animal testing.

#### 1.6.5. Disadvantages of 3D Cell Culture Systems as a Preclinical Models

Three-dimensional cell cultures offer an exciting substitute for animal use in agreement with the 3R principles. However, its improvement faces different challenges and its use can be limited because of certain drawbacks.

Use of animal-derived components: While tissue engineering aims to reduce animal use, a major constituent of cell culture media still uses components of animal origin—mainly serum. Animal serum, like fetal bovine serum (FBS), is a supplement in culture media to provide hormones and various growth factors for cell development [39]. However, in addition to raising concerns for animal welfare and biosafety, animal serum’s use decreases the reproducibility of experiments because of lot-to-lot variations [80,81,82]. Serum replacement will be discussed below.

High cost: 2D cell cultures show the potential to have relatively low costs, whereas advanced 3D culture systems are considered more expensive. However, price varies according to the 3D technique used. For some techniques, specialized equipment is required and contributes to raising expenses [83]. Therefore, different studies aim to develop and establish low-cost 3D cell culture techniques [84,85,86].

Time-consuming and low agreement for high throughput screening and automation: Some 3D cell culture techniques are technically demanding and time-consuming [84]. Therefore, they are not as suitable for high throughput screening as 2D systems [87,88,89]. Their automation also faces challenges. However, Aijian and collaborators have automated the hanging drop technique using digital microfluidics but at higher costs [90].

Variable reproducibility: The reproducibility of 3D appears to vary according to the technique used. For instance, scaffold-based systems seem to show poor reproducibility if animal-derived components are used [89], whereas scaffold-free methods are more successful [91,92]. For instance, the hanging drop method has shown nearly 100% reproducibility for numerous cell lines [91].

Lack of vasculature: The absence of vascularization in engineered tissue is a disadvantage for tissue transplants since it is a significant reason for poor graft take [93]. The development of endothelial capillaries within 3D engineered tissues could improve graft take [94,95] and contribute to creating a more relevant tumour microenvironment for cancer study models or studying angiogenic functions [78,96,97].

Lack of interaction with the immune system: A significant drawback of all in vitro models is that the created microenvironment does not include immunologic components, thus contributing to the gap between in vitro and in vivo models. The incorporation of immune cells and lymphatic capillaries to make cancer models more relevant is still a challenge but seems like a reachable objective [78,96].

Difficulty to produce relevant scaffolds and to mimic complex structures: Finding prefabricated scaffolds that match the structure and mechanical properties of native organ ECM can be a challenge [98], as is the case, for instance, for blood vessels [99]. Therefore, using scaffold-free methods, like the self-assembly method, could prove advantageous [89,98]. Moreover, there are difficulties regarding the capability of producing complex 3D structures using cell culture. Three-dimensional bioprinting has allowed the production of complex structured scaffolds, but it still has many limitations [100].

Because of its various methods and relatively recent emergence, 3D cell culture is still challenging to standardize. Achieving a low-cost 3D culture that combines high throughput, reproducibility, time efficiency, and versatility remains a challenge but could lead to a decrease in animal use in agreement with the 3R principles.

Compared characteristics of animal and 2D and 3D cell culture models are presented in Table 1.

## 2. Three-Dimensional Cell Culture Methods

Advances in biological research have led to developments in cell culture to improve the current models used for in vitro studies. The emerging way to cultivate cells in a 3D manner rapidly gained popularity due to the possibility of overcoming the limitations of the 2D monolayer cell culture. Nowadays, many methods exist to produce a specific tissue in 3D cultures from cells that can then be used for research, transplantation, or drug development. The advantages and disadvantages of each method described in this review are summarized in Table 2.

### 2.1. Scaffold Methods

The most common way to cultivate cells in TE is to use a scaffold. A scaffold is a 3D structure made of synthetic, natural, or mixed components, on which cells can be seeded to form a tissue. Once the cells have populated the scaffold and started producing their own ECM, the scaffold is often degraded into by-products. The properties of the scaffold, such as permeability, surface chemistry, porosity, and mechanical characteristics, as well as its degradation profile and its ability to release biomolecules, contribute to mimic the cell’s microenvironment [107,109]. The scaffold matrix closely interacts chemically and physically with the cells and directly influences their capacity to proliferate, differentiate, and secrete extracellular components [110]. Therefore, the choice of the matrix is heavily based on the cell type used and the nature of the study [111].

#### 2.1.1. Synthetic Scaffold

Synthetic scaffolds are artificially made from organic or inorganic components such as metals, ceramics, and synthetic polymers. Compared to natural scaffolds, these materials, by their constitution, often lack biocompatibility with cells, sometimes requiring the encapsulation of signaling molecules, growth factors, hormones, or cellular markers to their surface, to ensure better cell proliferation and differentiation [110,112,113]. However, they usually have excellent physical properties, well-characterized components, and low risk of pathogen transmission, making them useful tools for tissue regeneration of cartilage and bones, for instance [112,114,115].

##### Metals

Metal atoms have many exciting properties when combined into nanoparticles or porous materials. For instance, silver has toxic properties useful to limit unwanted contamination by microorganisms. Gold has good tunability, stability, and biocompatibility [116]. Finally, cobalt and nickel show magnetic properties when submitted to electromagnetic fields [117]. Palladium, titanium, and magnesium are also commonly used [118,119,120]. Many scaffolds using metal atoms have been developed over the past few years, improving their mimicking of the ECM and trying to overcome certain limitations such as oxidation and aggregation issues [118,121].

Porous metallic materials have a wide-spaced rigid structure with low density and high mechanical strength, closely resembling the bone in vivo. Their shape and permeability are ideal for cell infiltration of the scaffold with good diffusion of nutrients, oxygen, and metabolic waste [120,122]. Their porosity is highly tunable, affecting the stiffness and strength of the scaffold, the diffusion capacity, and the extent of the migration of the cells, playing with the density of the tissue generated [123,124]. Biodegradable porous metallic scaffolds can be made by selective laser melting [125], powder metallurgy [126], injection of inert gas into a melt [127], sintering of particles [128], vacuum foaming [129], investment casting [130], thixocasting [131], and other methods [122].

Metal nanoparticles can be combined with a polymer to form a polymer or hydrogel nanocomposite [132,133]. This type of scaffold prevents the nanoparticles from aggregating while preserving their properties. The polymer used can be of natural (polysaccharides, proteins) or synthetic (polymeric acids, polyvinyl alcohol) nature. The nanoparticles notably improve the mechanical properties, the thermal stability, and the electrical conductivity of the polymer matrix [134]. They also provide the scaffold with a broad-spectrum antibacterial activity [135]. Polymer nanocomposites are mainly made by melt mixing, by in situ polymerization of a monomer in the presence of metal nanoparticles, or by the in situ reduction of metal salts/complexes in a polymer matrix [136].

Novel approaches combine metals with natural ECM components to generate hybrid scaffolds with excellent biological properties while preserving good structural strength. For example, a team in California developed a new scaffold using a stainless-steel mesh coated with bovine and rat collagen that could be beneficial in cardiac and vascular membrane grafts [137].

##### Ceramics

Bioceramic-based scaffolds are mainly used in bone and dental TE [138,139]. The term bioceramic refers to a class of ceramic materials that have biocompatibility with the human body. They are made from stable components that can be either biologically active or inert [138]. These materials also have excellent physical stability, antibacterial effects, and antithrombus effects [140]. They have chemical structures that closely resemble the inorganic components of human bones [141]. They offer a 3D porous structure that enables an effective diffusion of nutrients and metabolic wastes, thereby allowing cell migration and ingrowth of the tissue from the periphery toward the inner center of the scaffold [142]. In particular, the porosity and the interconnection of the pores expand the surface area while creating roughness on the scaffold’s surfaces, promoting osteoblasts adhesion with the scaffold [143]. However, generating a ceramic scaffold with porosity and pore sizes as high as a human bone significantly decreases its structural strength [144,145].

Common ceramic materials used in TE are calcium phosphate, hydroxyapatite, tricalcium phosphate, biphasic calcium phosphate, calcium silicate, tricalcium silicate, and bioactive glasses. Particular ions, such as zinc, strontium, magnesium, and manganese, can also be incorporated into the scaffold, adding interesting properties to it and promoting better cell differentiation into osteoblasts [146]. Bioactive glasses are mainly composed of Na_2_O, CaO, SiO_2_, or P_2_O_5_ [147]. They possess an interesting osteoconductivity and osteoproductivity, favouring progenitor cells’ proliferation and differentiation [148]. In controversy, they have a significant brittleness that can cause the collapsing of the highly porous structure, thereby altering the tissue ingrowth [138].

Ceramic scaffolds can be made by gas foaming [149], freeze-drying [150], fibre bonding, particulate/salt leaching [151], emulsification [152], phase separation/inversion, and, more recently, 3D printing [138]. In general, scaffolds are designed to mimic as closely as possible the in vivo environment. Three-dimensional printing of ceramic scaffolds offers a tunability of many parameters (porosity, pore shape, pore interconnectivity, architecture) that cannot be achieved by more traditional fabrication methods [146].

##### Polymers

Polymers are an assembly of a repeating subunit and form interesting materials for TE. Synthetic polymeric scaffolds are commonly made of polyesters, polyether esters, polyurethanes, and synthetic silks [153]. Polymers can be massively produced with controlled composition and molecular weight with no variability between the different production lots [154]. They can be made by solvent casting [155], particulate leaching [156], electrospinning [157], emulsion freeze-drying [158], phase separation [159], melt molding [160], selective laser sintering [161], stereolithography [162], fused deposition modeling [163], and 3D printing [164,165]. The high tunability and reproducibility of polymers are ideal for scaffolds with specific properties [153]. A synthetic polymer can also be combined with a synthetic or natural polymer to generate a scaffold with complimentary properties from both materials [166].

A significant drawback of the use of synthetic polymeric scaffolds is that their matrix lacks cell recognition signals for proper cell adhesion to the scaffold, particularly compared to natural scaffolds [167]. Therefore, cell adhesion molecules, proteins, and growth factors must be added to the matrix to better mimic the cell microenvironment and to have a desirable tissue ingrowth [168]. Furthermore, some polymers such as poly (α-hydroxy esters) generate acidic by-products when they are degraded by the cells, changing the pH of the medium that diminishes cell proliferation and alters their survival. This pH change can also lead to an inflammatory response to the graft [169]. With that in consideration, synthetic polymeric scaffolds still have lower immunogenicity and a reduced risk of graft rejection because of their lack of bioactivity compared to natural scaffolds [170].

#### 2.1.2. Natural Scaffolds

Over the years, natural scaffolds have been preferred over synthetic scaffolds in TE because of their inherent biocompatibility [113]. In general, scaffolds are designed to closely mimic the cell microenvironment to enable proper cell proliferation and differentiation [154]. Natural scaffolds easily achieve this objective without requiring the encapsulation of cell signaling molecules and growth factors into the matrix [171]. Moreover, they have an excellent degradation profile where only non-toxic by-products are generated [113]. Natural scaffolds include polysaccharides, proteins, ECM-derived scaffolds, and acellular matrices. They are produced by living organisms such as bacteria, yeast, plants, and animals [172,173,174].

Consequently, unknown exact composition and in-between lots variations are some limitations of this cell culture method [103]. Natural materials are not always easily processed into the required shapes and sizes [113]. They also have a higher risk of pathogen transmission and graft immunogenicity than synthetic scaffolds [114,170].

##### Polysaccharides

Natural polymers are interesting candidates to form 3D scaffolds. Polysaccharides are a type of natural polymers commonly used in TE.

Chitin is a substance found in the composition of the exoskeleton of crustaceans and insects, or fungi. While this material lacks essential properties for TE, it can undergo deacetylation via chemical hydrolysis in alkaline conditions or by enzymatic hydrolysis, to generate a copolymer of randomly distributed *N*-acetyl d-glucosamine and d-glucosamine subunits [175,176]. Because of its interesting properties, this newly formed material called chitosan is widely used in various fields such as wound healing, bone tissue regeneration, and drug delivery [177]. Chitosan is slightly positive at acidic conditions due to its amino groups, which enable interactions with negatively charged molecules and membranes. It has good biocompatibility, degradation profile, and mucoadhesivity, and is non-toxic and non-antigenic. It also has antibacterial properties as well as high adsorption properties [176,178]. However, this material has weak physical strength. This limitation can be overcome by combining chitosan to materials with better mechanical properties like calcium phosphate, hydroxyapatite, or silk [179]. In industry, chitosan is mainly produced by fungi since it offers a more controlled and scalable production with less lots of variation [177].

Alginate is a primary component of cell walls of brown seaweed and bacteria [180,181]. This polysaccharide’s subunits are β-d-mannuronic acid and α-l-guluronic acid through a (1→4) linkage, which confers a pH-dependent negative charge to the material. Playing with the subunits ratio gives tunability to alginate for mechanical and biological properties [179,182]. This low-cost material is hydrophilic, biocompatible, has a good degradation profile into non-toxic by-products, is non-antigenic, and can be chelated with cations into microspheres [183,184]. Alginate can be turned into hydrogels and porous scaffolds by various crosslinking methods such as phase transition and free radical polymerization or freeze-drying and electrospinning [181]. It is easy to manipulate and to form scaffolds of different shapes and sizes with it. On the downside, alginate, by its composition, lacks cell recognition patterns, sometimes resulting in lower cell adhesion. Therefore, cell receptors and signaling molecules into the matrix might be required [185]. Recently, new advances have been made to produce a new form of cellular building blocks that can be self-assembled into complex tissue constructs. Tissue strands (Figure 4) are made by injecting a cell suspension through a semi-permeable tubular alginate capsule. These capsules are made with a coaxial nozzle apparatus and have low variability in shapes and sizes. Once the cells are injected into them, these capsules have their end tied using vessel clips. Following a five-day culture, the alginate is then dissolved with a 1% citrate solution, leaving pure cellular strands. These tissue strands can then be cut into smaller pieces to be assembled or be used as bioink to form larger tissues [186,187]. During the deposition of the cell pellet, alginate porogens can be added to the cellular suspension to form porous tissue strands. This allows for a better diffusion of nutrients and metabolic wastes and enhances cell survival [188].

Hyaluronic acid is a natural linear polysaccharide found amongst others in conjunctive, epithelial, and nervous tissues of vertebrates. It is composed of repeating *N*-acetylglucosamine and d-glucuronic acid units and has an anionic charge at physiological pH [189,190]. This material is biocompatible since it is a primary component of the ECM, has a characteristic viscosity and elasticity, and good biodegradability. Mainly formed into hydrogels, it is known to induce osteogenic differentiation and regulate bone tissue formation [191]. Hyaluronic acid and its derivatives can be chemically modified with ease to gain desired functions and have low immunogenicity [190,192].

However, they have poor mechanical properties and can be hard to manipulate into desired scaffolds [102]. Even though hyaluronic acid originates from animals, it is nowadays produced by microbial fermentation [193].

Cellulose is commonly found in bacteria, tunicates, and plants, a primary component of their cell walls. It is formed of repeating β-d-glucose with a β(1→4) link [194]. Depending on its origin, cellulose might have slightly different properties when shaped into scaffolds [195]. In general, this material has excellent biocompatibility and mechanical strength, is hydrophilic, has practical optical transparency, and is inexpensive. Other polymers can be mixed with cellulose to add specific properties to the scaffold [196]. It also enables proper cell adhesion, proliferation, and osteogenic differentiation. Thus, it is a good candidate for bone tissue reconstruction and medical implants [179]. A particular limitation to cellulose is that it has a low degradation rate in vivo, which can be problematic for tissue transplants [102].

##### Proteins

Natural polymers can also be made of proteins and peptide strands that can be shaped into scaffolds for 3D cell cultures.

As one of the ECM’s primary components, collagen is the most abundant protein in humans and is responsible for maintaining structural integrity in various tissues [197,198]. It, therefore, has inherent biocompatibility and low antigenicity [199]. Collagens are a group of triple α- helix proteins. Type I, II, III, and V are the ones most found in the ECM with the ratio depending on the tissue type [200]. Collagen fibres are highly tunable and can easily be shaped into porous scaffolds or hydrogels [179]. They provide excellent support for cell proliferation and are degraded by cell enzymes. Cells can attach to the matrix through their integrins, thereby activating cell survival pathways [201]. On the downside, collagen scaffolds generally have low mechanical strength [202,203]. The primary production method is the extraction of collagen-rich tissues such as skin, tendons, intestine, and bladder from animals or human sources. The collagen fibres are then collected through a purification process of the tissue [204]. Even though collagen is an excellent candidate for 3D scaffolding, the necessity to use animals for its production is not in harmony with the 3R principles. Nevertheless, new options have emerged, such as recombinant human collagen produced in tobacco plants at a low cost [205].

Gelatin is a polymer directly derived from type I collagen that has undergone irreversible hydrolyzation through heat and enzymatic processes. It is a mix of proteins, mineral salts, and water [206]. This material has gained interest in TE because of its biocompatibility, biodegradability, and antigenicity. It promotes cell adhesion and is usually non-toxic, although this property may vary depending on the crosslinking reagent used [207,208,209]. Due to its weak mechanical properties when turned into a scaffold, it is commonly mixed with synthetic polymers to improve this parameter [210]. Gelatin is extracted from various animal sources such as cows, sheep, pigs, fish, and even insects [206]. Like collagen, this material conflicts with the 3R principles.

Fibrin is a natural protein playing an essential role in wound healing in the human body. Following an injury, fibrin aggregates serve as a structure for cells to adhere to and secrete new ECM components [211,212]. Fibrin is produced upon the interaction of fibrinogen with thrombin, as part of the coagulation cascade [213]. Therefore, fibrin scaffolds have inherent biocompatibility with high cell affinity and rapid degradability [214]. The scaffold made from fibrin fibres closely mimics the ECM and promotes cell proliferation, especially when mixed with growth factors and endogenous molecules [215]. The properties of this material are highly tunable by varying the initial concentrations of fibrinogen and thrombin. However, fibrin might require to be mixed with another polymer to overcome the weak mechanical properties and the rapid degradation rate of this material [216]. In an autologous context, fibrin can be generated from fibrinogen and thrombin extracted from the blood of the patient [217].

Silk is naturally made by certain Lepidoptera larvae, some arachnids, and a few flies [218]. It is composed of two fibroin fibres linked together by a triangle cross-section made of sericin proteins [219]. Its composition may vary depending on its origin. To be used as a scaffolding material, the sericin part of the silk needs to be removed [220]. Silk fibroins are highly biocompatible, have excellent mechanical properties, and have a good degradation profile where their by-products can be absorbed [219]. This material can be easily manipulated, and endogenous molecules can be encapsulated. Production of silk fibres is mainly made by extraction from lepidopteron insects of Bombycidae or Saturniidae families [218], but recombinant proteins are emerging [221].

As part of our alimentation and gaining interest because of its benefits on our health, it was recently reported that soy proteins, extracted from soybeans, were considered a potential candidate to replace animal-derived ECM proteins as scaffolding materials [222]. They have excellent biocompatibility and generate non-toxic by-products when degraded. This low-cost material can be turned into porous scaffolds or hydrogels and induce cell proliferation and survival [223,224]. Soy proteins, due to their low molecular weight, are highly prone to rapid enzymatic degradation. To overcome this limitation, mixing with synthetic polymers or crosslinking reagents is a way to improve soy scaffold’s mechanical properties. However, some crosslinking reagents are toxic to the cells, while synthetic polymers might have unwanted properties [222,225]. The use of soy protein scaffolds for TE still requires some improvements, but it has great potential, since it is a good example of an alternative solution to reduce animal use in research.

##### ECM-Derived Scaffold

Some scaffolding materials can be directly extracted from the cellular microenvironment. This is the case for Matrigel^®^, a material made of ECM components of Englebreth–Holm–Swarm mouse tumours [226]. It contains structural proteins such as collagen, laminin, and enactin, as well as various growth factors [227]. Since Matrigel^®^ is naturally produced and extracted through a relatively minimalistic procedure, its physical and chemical properties can be maintained to accurately mimic the ECM [228,229]. Mainly used as a hydrogel, it supports cell survival and proliferation [104]. This mixture of ECM components has been proven useful in various domains, such as cancer and stem cell research [227]. By an unknown mechanism, Matrigel^®^ helps maintain the pluripotency of stem cells in culture [227]. Since it is extracted from tumours in an animal, in-between lots variations and uncertain composition and concentration of components are a significant issue with this material [103]. Once considered as the gold standard scaffolding material in TE, this product is furthermore based on the inoculation and development of cancer tumours in mice for ulterior sacrifice and collection of extracellular matrices. This procedure is derogatory to the 3R principles, and alternative culture methods should be favoured.

##### Acellular Matrices

Tissues taken from living sources can undergo a decellularization process to remove all cells from its ECM. The resulting matrix is made of structural proteins such as collagen and glycosaminoglycans, and endogenous molecules [230]. In various domains such as cardiac surgery and breast reconstruction, it is used as scaffolding material. It provides an authentic physiological microenvironment for cells to grow into a specific tissue from which the scaffold is derived [231,232]. Commercially available acellular matrices can also be a synthetic mixture of many ECM components from different tissue sources [230].

Since acellular matrices can be extracted from a large variety of tissues with different properties, many decellularization protocols have been developed throughout the years [233]. Decellularization is possible by a chemical, biological, or physical approach, or a combination of the three. Chemical decellularization includes the use of detergents and alcohols, acid-base reactions, and hypotonic–hypertonic reactions. The biological approach is based on the use of enzymes such as collagenase, trypsin, or nuclease to remove all cells. Decellularization by physical methods includes mechanical abrasion, electroporation, and temperature and pressure treatment [230]. The objective is to remove all cellular components while preserving, as much as possible, the ECM components and structure for their excellent biomimicry properties.

Acellular matrices as bioscaffolding materials have significant interactions with the cells for proper survival, proliferation, and differentiation. Since collagen, amongst other ECM components, is highly preserved across species, they have high biocompatibility and generally do not generate a significant immune reaction in the host [234,235]. However, the complete decellularization of the tissue requires an intensive protocol that sometimes denatures ECM components and alters its structural properties. This results in variation between the scaffold and the in vivo ECM that can affect cell behaviour and limit cellular colonization [231]. Furthermore, the sterilization of the acellular matrix scaffold can be difficult or incomplete. Simple sterilization treatments like incubation with solvents do not provide sufficient penetration to the core of the scaffold. More drastic methods, like gamma irradiation and ethylene oxide exposure, affect the structural and mechanical properties of the scaffold [233].

With current techniques and scaffolding materials, acellular matrices are the ones that undoubtedly better mimic the microenvironment of the cells. However, these materials originate from tissues taken from living organisms. While this method is interesting in an autologous context, the use of animal tissues for cell culture purposes does not respect the 3R principles.

Regarding the 3R principles, not all scaffolding materials that contribute to reducing animal use in TE. Even though synthetic materials are made artificially without involving living organisms, they have limited biological properties compared to natural materials. While microorganisms make some natural materials, some of them still require tissues from animal sources to be formed. Therefore, methods respecting the 3R principles should be privileged, and improvements should be made to overcome their limitations.

#### 2.1.3. Hydrogels

Hydrogels are made from natural or synthetic hydrophilic polymers or a mix of both types such as poly(ethylene oxide), poly(vinyl alcohol), polypeptides, agarose, or gelatin. The difference with previously described scaffolds resides in their water content. In the case of hydrogels, water is up to 95% of the volume [236,237]. This propriety has the distinct advantage to enable easy access to nutrients, growth factors, and oxygen to cells proliferating through the scaffold [238]. It also improves the biocompatibility of the scaffold [239]. Hydrogels are highly flexible while having adequate structural stability [104,106]. Their water retention and their soft rubberish surfaces resemble human tissues [239].

These scaffolds have an excellent tunability for 3D cell culture [107]. For instance, their degradation rate can be easily modified by changing the number of crosslinks between polymers [240]. However, they have some limitations. Since endogenous factors are not embedded directly into the matrix, an uneven gradient of soluble factors can alter proper cell differentiation. The particular architecture of the scaffold also makes it harder to do cell imaging and analysis than other types of scaffolds: Cells can have limited accessibility for immunostaining. In addition, light scattering, refraction, and attenuation occur in 3D composite cell-laden gel [241].

Hydrogels can be made from physical or chemical crosslinking of polymers [242]. Nowadays, common fabrication methods are 3D printing [243], layer-by-layer fabrication [244], microfluidic-based fabrication [245], and self-assembly [242].

### 2.2. Scaffold-Free Methods

The first method developed to cultivate cells without any external structural support was the hanging drop technique and was introduced by Johannes Holtfreter in 1944. Initially used for the culture of embryonic stem cells, it is nowadays primarily used to generate spheroids [246]. This method opened a new path of techniques that do not rely on the scaffold to form a tissue of interest. Since the introduction of the hanging drop technique, the term “scaffold-free” has been given various definitions and covers a large number of techniques (Figure 5).

#### 2.2.1. Principles

In vivo, stromal cells, mainly fibroblasts, are surrounded by structural proteins such as collagen and laminin, and endogenous molecules secreted by the cells. This microenvironment composition varies depending on the tissue source and localization in the organism. Interacting with the cells through signaling pathways affects their growth, differentiation, migration, and apoptosis, thereby regulating the tissue properties and specificity [247,248]. When cultured in vitro under optimal conditions, cells can produce and deposit their matrix components to form an ECM similar to the one found in vivo. The scaffold-free methods rely on this ability to generate a specific tissue from the self-aggregation of cells producing their 3D matrix [96,110,249].

#### 2.2.2. Spheroid

Spheroids are spherical 3D cell aggregates in suspension [249]. They were first developed by Sutherland and coworkers in 1970 to recapitulate human tumours’ functional phenotype [250,251]. Due to their 3D structure, outer layer cells are directly in contact with the culture medium, while cells in the inner core only obtain nutrients and oxygen by diffusion. This results in a proliferative cell activity on the perimeter of the spheroid and a hypoxic central zone with quiescent and necrotic state cells, particularly in large spheroids where diffusion is inefficient. This cellular heterogeneity contributes to mimic cancer tumours in vivo [53,103] closely. Spheroids can be made from one or many cell types and can be of various sizes, depending on the initial cell deposit [110]. They have many great utilities ranging from cancer invasion profiling to drug screening [77,252]. Spheroids are mainly made of four methods.

##### Hanging Drop Technique

The hanging drop technique, like suggested by its name, is done by inverting an upside-down Petri dish top cover containing droplets of cell suspension. The bottom of the Petri dish contains sterilized water to maintain a specific humidity level [253]. By microgravity, all the cells go at the bottom of the drop to the air/liquid interface [254]. Without adhesion to a surface, cells aggregate themselves to form a dense sphere [91,249]. A maximum of 50 μL of cell suspension can be deposited onto the cover. Otherwise, the surface tension is too weak to maintain the drop in place [255]. The spheroid size is controlled by the initial number of cells suspended. The co-suspension or consecutive addition of multiple cell types allows the formation of multicellular spheroids with distinct cell layers [110]. The hanging drop method is simple to use and highly reproducible with the formation of one spheroid per drop with low parameter variability [91]. Since this method is not suitable for frequent media change, large spheroids might require a transfer to a propagation plate with higher media volume to ensure proper culture conditions [110].

##### Low-Attachment Plates

An alternative method to produce spheroids is by using low-attachment plates. Often made of polystyrene treated with hydrophilic or hydrophobic coatings preventing cellular attachment, their wells are suited for producing a single spheroid [256]. By containing many wells and having a large volume, low-attachment plates are frequently used for multicellular spheroids culture in cancer research [110]. The cell suspension is seeded into individual wells and is incubated. Cell media can be changed frequently with automated instruments [257].

##### Magnetic Levitation

The magnetic levitation method is an upgrade from the low-attachment plate method to produce spheroids. It relies on the use of magnetic nanoparticles. Before being seeded into the wells of a low-attachment plate, the cells are mixed with magnetic particles. During the culture, a magnetic field is applied to elevate the cells toward the air/liquid interface, promoting contact between cells and their differentiation to form a dense aggregate [110]. This method is well suited for co-culturing multiple cell types for the production of heterogenic spheroids [258,259].

##### Bioreactor

Another method for producing spheroids is by using a bioreactor. Bioreactors can be classified into four categories: spinner flask bioreactors, rotational culture systems, perfusion bioreactors, and mechanical force systems [260,261]. The general principle for the formation of spheroids is the same for all the types of bioreactors: a cell suspension with optimal density is filled into a chamber that is constantly shaken, either by stirring, rotating, or perfusing the culture media through a pumping system. This movement allows the cells to aggregate. Bioreactors also have flowing systems to generate movement of nutrients and metabolic wastes and maintain homogeneity of the physical and chemical factors within the chamber. This method is made for large scale production, although a significant variability is observed between the spheroids. Moreover, it does not allow any control over the spheroid size and cell number [106,262].

#### 2.2.3. Organoids

The term “organoid” means self-renewing 3D cultures derived from primary tissue, embryonic stem cells, or induced pluripotent stem cells (iPSCs) that have a similar organization and functionality as the tissue/organ from which the cells originate or to which they are differentiated to, in the case of iPSCs [110]. Organoids can be classified depending on if they are formed from tissue or stem cells. Tissue organoids are mostly comprised of epithelial cells with no stroma. Stem cells organoids are made of embryonic stem cells or induced pluripotent stem cells or primary stem cells extracted from neonatal tissue or specific organs [107]. This type of culture is based on stem cells’ ability to self-organize in vitro into a tissue sharing the structural and functional properties of the organ or tissue from which the cells originate [263]. Organoids have been used to mimic organs such as the brain [264], retina [265], stomach [266], lungs [267], thyroid [268], small intestine [269], liver [270], kidneys [271], and many more. They can be used in various domains, such as human pathologies and cancer research [263]. Many methods are used to produce organoids. Cells can be directly cultured into a 2D monolayer on a bed of feeder cells or an ECM-coated scaffold to form the organoids after cell differentiation. Cell cultures can also be supported mechanically to allow further differentiation of primary tissues. The hanging drop or the low-attachment plate techniques can be used to generate embryoid bodies. Finally, the serum-free floating culture of embryoid body-like aggregates can be made with quick reaggregation in low-attachment plates [107].

#### 2.2.4. Microfluidic

Microfluidic systems are an improvement of the scaffold-free methods for a better mimicry of a living organism. They provide a continuous supply of nutrients and oxygen while removing metabolic wastes, creating an artificial network. This enables the production of large engineered tissues and the assembly of multiples organoids or spheroids to generate in vitro a whole system [103,272]. Microfluidic systems also allow precise culture conditions and better monitoring of the cells [273].

#### 2.2.5. Self-Assembly Method

In 1972, B.R. Switzer and G.K. Summer showed that ascorbate, an enzymatic cofactor of lysyl- and prolyl-hydroxylase, stimulates the production of collagen type I by human dermal fibroblasts when added into the culture media [274]. Indeed, ascorbic acid facilitates the proline’s hydroxylation, which is crucial for collagen fibrils stability and their deposition [96]. This was followed by a groundbreaking experiment done by R. Hata and H. Senoo in 1989. They demonstrated that dermal fibroblasts could deposit and organize enough ECM components within a few days of culture to create a 3D stromal sheet that can be stacked or rolled to form a 3D organ substitute [275]. This led to a whole new way of producing engineered tissues without requiring external support (Figure 6).

##### Protocol

To generate a complex human tissue construct, cells firstly need to be extracted and isolated from a patient/volunteer’s biopsy. The biopsy is cleaned with a sterile saline solution containing antibiotics. The stroma and the epithelium are enzymatically separated using thermolysin. Then, collagenase can be added to the stroma to extract mesenchymal cells. Trypsin can be used to extract epithelial cells from the epithelium. By adding 50 μg/mL of ascorbic acid into the media throughout the mesenchymal cells culture time, these cells produce and assemble their own ECM components [276]. This enables the formation of stromal sheets that can be collected by forceps and stacked to be used as cellularized scaffolds. Instead of being stacked, stroma cells can also be reseeded onto a stromal sheet, resulting in a more thick and solid cellular sheet. Then, epithelial cells can be seeded onto this assembled stroma. After a week in culture to allow a full cover of the apical region by the epithelial cells, the cellular sheets are maintained for 21 days at an air/liquid interface to induce the maturation of the epithelium cells. Thereby, engineered cellular assembly mimicking the native tissues is obtained and can be used as models for research or as grafts for autologous transplants [96,108].

##### Current Tissues Produced Using the Self-Assembly Technique

Several tissues have been obtained using the self-assembly method developed at LOEX by Dr. François A. Auger and coworkers (Figure 7 and Table 3). These tissues can be used for pathological and pharmaceutical research, and can be transplanted in patients. Skin, blood vessels, corneas, urological tissues, adipose tissue, bone, and vaginal mucosa can be made using the self-assembly method. Skin tissue is obtained by stacking the dermal fibroblast-derived stromal sheet after four weeks of culture seeded with keratinocytes that form the epidermis [276]. The self-assembly model of skin is currently used to help wound healing of severely burned victims [277,278]. Corneal tissues can be obtained using epithelial, stromal (keratocytes), and corneal endothelial cells [279]. Bladder and urethra tissues could be made from skin fibroblast or bladder mesenchymal cells, for the stromal compartment, combined with urothelial cells, for the epithelial compartment [280,281,282]. Adipose stromal tissue can be produced with a self-assembly method using adipose-derived stem cells (ASCs) [283]. ASCs have osteogenic potential. In recent years, a thin bone substitute has been produced using osteocytes with the self-assembly method [284]. ASCs also demonstrated the potential to produce blood vessels and treat scars; they have also been used to improve bladder tissues [285,286,287]. Vaginal mucosa could also be made with the self-assembly method using organ-specific cells [288].

## 3. Applications of 3D Cell Cultures

### 3.1. Pathology

Various models to study specific pathologies have been developed using the self-assembly approach, thus contributing to enhance personalized medicine and reduce animal testing.

For instance, 3D in vitro psoriatic skin models that successfully demonstrate psoriasis characteristics, like excessive proliferation and abnormal differentiation of keratinocytes, have been developed by Jean and collaborators using cells derived from patients with psoriasis [71]. The model, made by the self-assembly method, has been used in studies that aim at better defining cellular events of the pathology or characterize specific distinctions of psoriatic skin [65,72]. Skin fibrosis was also a selected pathology to be investigated using the self-assembly protocols, especially hypertrophic scars [289] and scleroderma [290,291]. The same protocol was used to study amyotrophic lateral sclerosis (ALS). Those constructs made from ALS patients’ biopsies led to the identification of disease-specific biomarkers that can potentially be used as an early diagnosis tool of the disease [74]. In addition, various self-assembled 3D stromal tissues have been successfully used to study cancers [96]. The tumour microenvironments of skin and bladder cancers have been replicated adequately by this approach [63,64]. The use of these two cancer models for drug testing shows the potential to reduce the need for animal testing. The bladder mucosa model has also been used to study cystitis induced by ketamine use [292].

### 3.2. Pharmacology

The road to the development of a new drug is very long and expensive. It takes approximately 13.5 years to develop and market a drug and costs 1.8 billion US. However, only 7 to 11% of the developed drugs will be commercialized [5]. Animal models were used and are still being used today for drug development. For pharmaceutical research, 2D and 3D cell cultures represent an attractive alternative to animal models. Besides the advantages presented above, cell cultures offer the possibility to develop a personalized drug by using the patient’s cells. Personalized therapy is beneficial for cancer patients since each tumour is different [293].

Cell culture is also beneficial for high-throughput screening (HTS) in drug discovery and characterization. HTS is a process developed in the 1990s from the fruitful wedding of combinatory chemistry and automation of techniques [294]. The goal of HTS is to screen a library containing a large number of potentially active molecules against a specific target to find the best potential drug. HTS could be used to discover a new drug or characterize its toxicology or metabolism [295]. With cell cultures, absorption, distribution, metabolism, excretion, and toxicity of a drug could be elucidated at the beginning of drug development and not in preclinical, i.e., on animals, or in clinical trials [294]. Cell-based HTS could reduce the number of animals used in research and reduce the cost and time of the development of a drug.

Drugs can be tested on 2D cell cultures, but monolayer cell culture has some limitations, as presented above. One of the significant flaws in the 2D cell culture is that it could not reflect the drug’s effect on the architecture of the tissue and cell–cell adhesion [107]. Indeed, cells in 3D cultures produce extracellular matrix (ECM), which is not produced by cells in 2D cells. ECM contains many components, such as growth factors, glycosaminoglycans, proteoglycans, and matrix proteins [110]. These factors regulate important cellular mechanisms such as cell proliferation, cell adhesion, cytoskeletal rearrangement, and many more [110]. A drug could interact with ECM components and modify some cell mechanisms. Interaction of the drug with ECM components would not be similar on a 2D than a 3D cell culture because there are not the same ECM components in a 2D cell culture.

On the other hand, these interactions could be studied on 3D cell cultures. Cells do not interact only with ECM in vitro, but they also interact with stromal cells such as fibroblast, adipocyte, and glial cells [110]. Interactions between cells can influence the drug response in vitro. For example, the interaction between the tumour and the stromal cell is the cause of many cases of drug resistance [296]. Once again, this phenomenon would not be observed in 2D cell cultures but could be in 3D cell cultures, as 3D cell culture reproduces more accurately the environment of the native tissue [110].

In recent years, many studies have been done on drug discovery and drug response for cancer treatment using 3D cell culture. For example, in 2018, a team from Columbia University tested drug response on organoids made from bladder cancer tumours. They isolated 16 tumour cell lines from individual patients, and 40 compounds were tested on nine organoids cell lines, and ten more compounds were tested on nine cell lines. These compounds contained molecules as trametinib, Erlotinib, and Ganetespid. Different drug responses were obtained with the drug screening depending on the cell lines. Some of these potential molecules were tested on xenograft in mice to validate drug response, and the same response was obtained in organoids and in vivo [297]. This method allows screening of components to find the most convenient drug for patients. Furthermore, instead, to test a large number of compounds on mice, only the ones that were promising on the organoid drug screening were tested on the animal model. This technique could reduce the number of mice that are used for drug testing, and maybe, in the future, if 3D cell culture improvements are sufficient, the step of testing the drug on an animal could be eliminated.

Nevertheless, 3D cell cultures in pharmaceutical research have some limitations. First, the experiments can be difficult to reproduce due to the heterogeneity of the cell population from the patient, but several works are in progress to improve this point. In addition, 3D cell culture is more expensive when compared with 2D cell cultures [53].

## 4. Perspectives

### 4.1. Serum-Free Medium

As an improvement of the current 3D culture methods, more research has been made to eliminate the use of serum in the culture medium. Fetal bovine serum (FBS) has been used for a long time as the ultimate supplement in culture media of human and animal cells due to its ubiquity. It contains various essential components, such as hormones, vitamins, proteins, growth factors, and traces of elements to sustain proper cell proliferation. FBS is extracted from the blood harvested by a syringe placed through the beating heart of cow fetuses at any developmental stage of the last two-thirds of gestation. These fetuses are mostly discovered by accident during the slaughter of pregnant cows. Developing a serum-free medium with the same properties as FBS would reduce animal use in research, as dictated by the 3R principles.

Many issues can be observed with the use of FBS in cell culture. First, it was shown that cow fetuses experienced pain and distress during the harvesting of their blood for FBS production, which raises ethical concerns [298]. Moreover, the exact composition of FBS is unknown and varies from lot-to-lot. This can lead to impaired cell differentiation and low experimental reproducibility [108]. Moreover, unintended interactions with tested substances are a possible risk. Safety concerns have been raised for laboratory personnel in terms of a possible exposition to endotoxins, mycoplasma, viral contaminants, and prions proteins [298]. Finally, FBS is an expensive product. Developing a cheaper alternative could be beneficial to lower experiment costs, particularly for small laboratories.

A universal synthetic medium replacing FBS has not been developed so far, but progress has been made in developing a specific serum-free differentiation medium. For instance, A.C. Volz and P.J. Kluger have achieved the differentiation of human adipose tissue-derived stem cells into adipocytes using a serum-free medium [299]. M. Fakhr Taha and collaborators have differentiated adipose tissue-derived stem cells into dopaminergic neurons in culture with a serum-free medium, with better results than their conventional low-serum medium [300].

### 4.2. Immune System

Recently, to improve 3D models to be used to replace animal models, several efforts have been made to include immune components into these models. Immunocompetent 3D skin models have been described [301] and recently, a psoriatic skin model has been refined by the addition of T cells [302]. Three-dimensional immunocompetent organ-on-chip models have also been explored to produce a more accurate model, at least in part, to replace animal models [303].

### 4.3. Connection between Organs through Vascularization

As was done for microfluidics [272], the connection between reconstructed tissues through a vascular system can mimic the interaction between organs. Blood vessels can easily be reconstructed using the self-assembly method [304]. These vessels can be connected with the pseudo-capillary network present inside the tissue, as it does after tissue-grafting in vivo [305,306,307,308]. Using several different techniques, functional three-dimensional tissues with perfusable blood vessels can be fabricated in vitro [309,310,311].

### 4.4. Combination of Techniques of Production and Maturation

Recently, the combination of several techniques of production or maturation, such as induced pluripotent stem cells, organoids, bioprinting, composite hydrogels, organ-on-chip, microphysiological systems, mechanical stimuli, innervation etc., (e.g., [312,313,314,315,316]), gave us the opportunity to produce a large spectra of complex organs. These organs could also be used for various applications such as potential transplantation (e.g., [317]) or disease modeling (e.g., [318,319,320]). For example, after demonstrating the functionality of hepatic cells bioprinted on collagen gels to drug screening [321], a mini-liver was produced using iPSC which differentiated into hepatocyte and self-organized into acini [322]. The basis of the production of liver substitute for transplantation has been laid: such a tissue required not only the right differentiation and organization of hepatocyte but also irrigation by a vascular network and potential reconnection to the host [317]. Another example: a gut-on-chip platform has been established to study various physiological aspect of this complex organ which is the gastrointestinal tract [323].

## 5. Conclusions

Even though all the methods used in TE to build tissue constructs were not primarily developed for research purposes, these techniques, by their various properties, allow scientists to produce reliable models of human tissues, organs, and pathologies. By their accuracy, these models significantly reduce the need to use animal models as research tools, thereby respecting the 3R principles. Depending on the method used, some improvements still need to be made to eliminate all use of animal-derived materials in TE, but 3D cultures can be considered viable ethical alternatives.

## Figures and Tables

**Figure 1 bioengineering-07-00115-f001:**
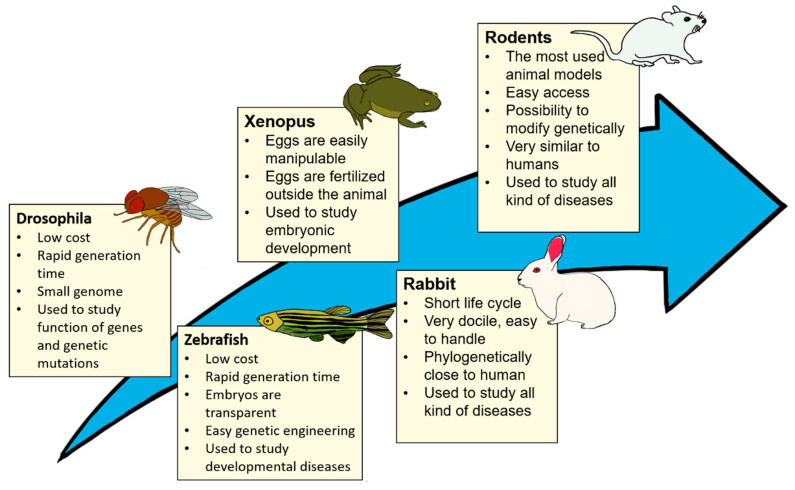
Advantages and applications of common animal models: drosophila [7], zebrafish [8], *Xenopus* [9], rabbit [10], or rodent [11] models in biomedical research.

**Figure 2 bioengineering-07-00115-f002:**
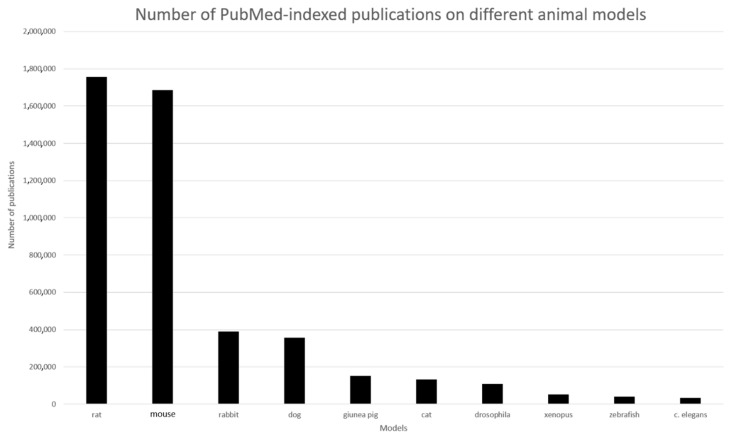
Number of Pubmed indexed publications on different animal models. The number of publications is the number obtained when entering the keyword in the search bar on PubMed website.

**Figure 3 bioengineering-07-00115-f003:**
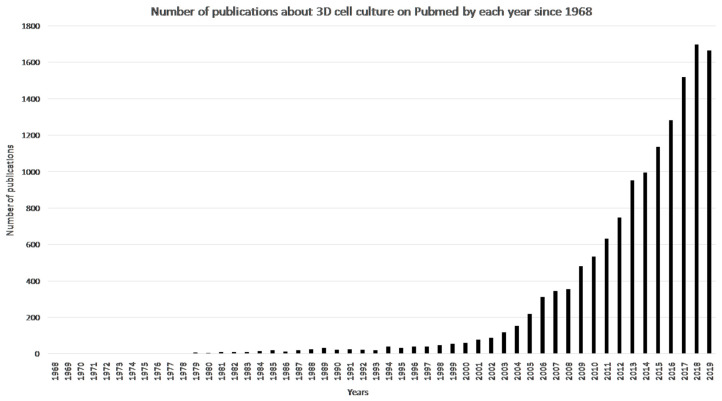
Number of publications per year about 3D cell culture on Pubmed since 1968. The number of publications is the number obtained when entering the keyword “3D cell cultures” in the search bar and applying the filter of results by year on Pubmed website.

**Figure 4 bioengineering-07-00115-f004:**
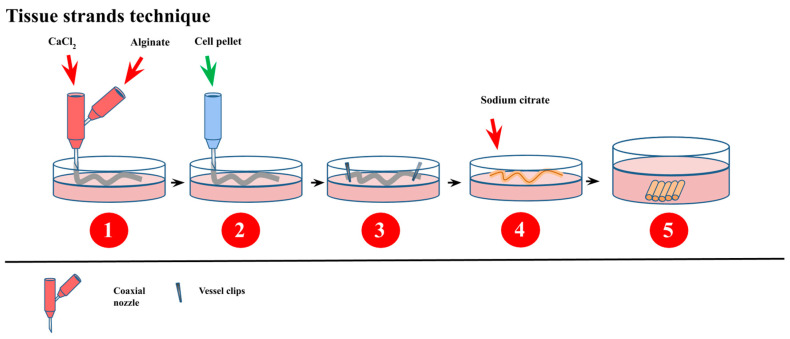
Schematic description of the tissue strands technique based on the ability of the cells to self-assemble into a tissue. Sodium alginate and a crosslinker solution are used to form tubular alginate capsules with a coaxial nozzle into a cell culture dish. Once polymerized, the capsules are filled with a cell pellet, and their ends are tightly sealed with vessel clips. After a 5–7 day culture, a sodium citrate solution is added to depolymerize the capsules, leaving complete and dense tissue strands that can later be used as building blocks to form complex larger tissues [186].

**Figure 5 bioengineering-07-00115-f005:**
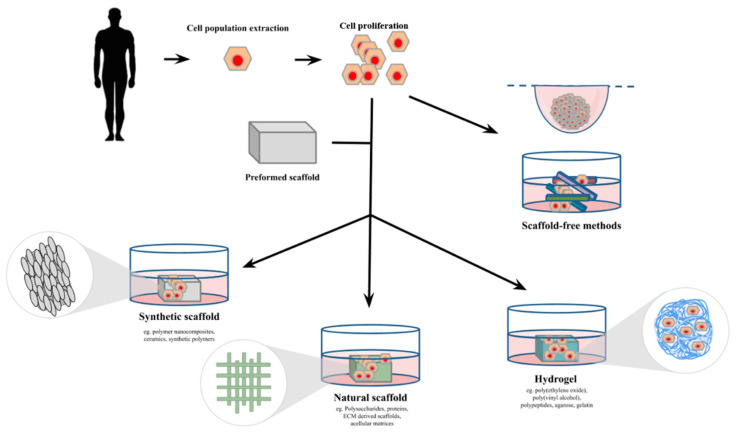
Scaffold vs. scaffold-free 3D cell culture.

**Figure 6 bioengineering-07-00115-f006:**
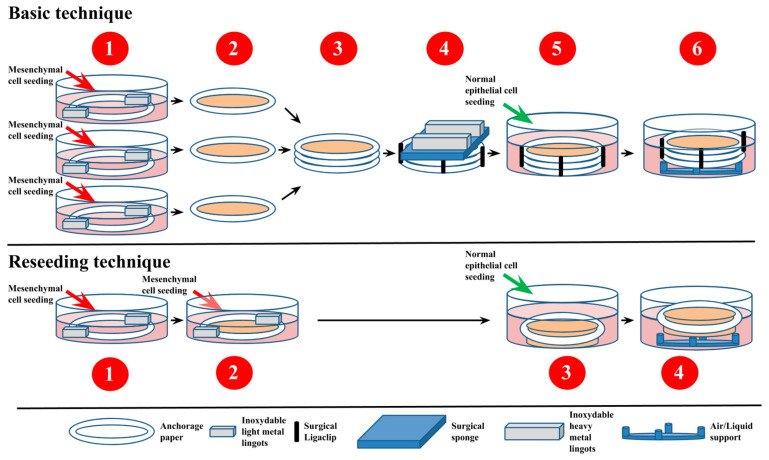
Schematic description of the basic and the reseeding self-assembly techniques used in Tissue Engineering. For the basic self-assembly technique, mesenchymal cells are seeded into three cell culture dishes containing a paper anchor and rust-resistant light metal weights at the bottom. After 28 days in culture with ascorbate, the stromal sheets formed are stacked upon each other for a variable time with a mechanical load composed of a surgical sponge and rust-resistant heavy metal ingots and with surgical Ligaclips to ensure the fusion of the sheets. Epithelial cells are seeded on top of the construct, and the culture is continued for seven more days. After that, tissues are mounted onto supports and maintained at the air/liquid interface for 21 consecutive days to ensure a complete maturation of the epithelium. The first step for the reseeding self-assembly technique is similar to the one for the classic self-assembly technique. However, instead of stacking the stroma sheets later, mesenchymal cells are reseeded on top of the stroma sheets after 14 days of culture with ascorbate. Fourteen days later, epithelial cells are seeded on top of the stroma sheets, and the culture is continued for an additional seven days. The tissue constructs are then maintained at the air/liquid interface with support for 21 days to ensure a complete maturation of the epithelium [108].

**Figure 7 bioengineering-07-00115-f007:**
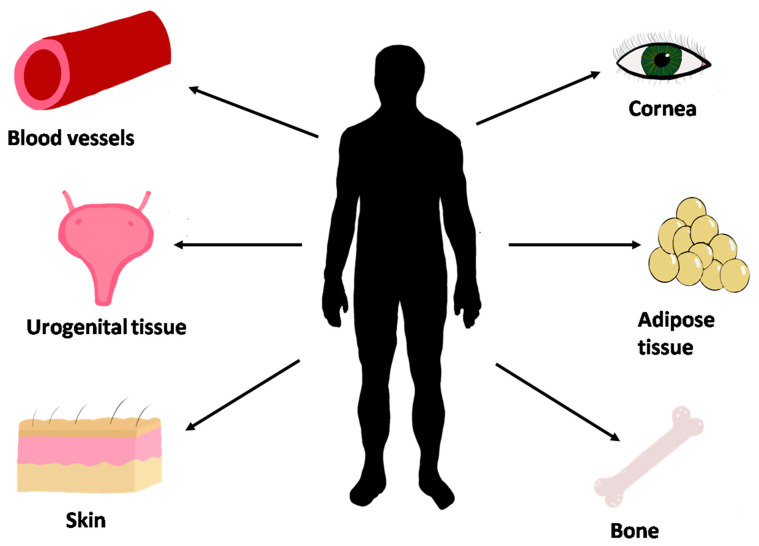
Types of tissues that can be reconstructed by tissue engineering using the self-assembly technique developed at LOEX.

**Table 1 bioengineering-07-00115-t001:** Characteristics of animal models compared to 2D and 3D cell culture systems.

	Animal Models	2D Cell Culture	3D Cell Culture
Cost	Relatively high expenses are relied on for the purchase of animals but also for housing and breeding, in addition to being time consuming.	Has potential for lower costs than in vivo experiments and involves relatively simple manipulations.	Often more expensive than 2D cell cultures. Some techniques can be technically demanding and time consuming.
Ethical concerns regarding animal welfare	Many ethical concerns are involved in animal testing because of the pain and distress experienced by models in certain protocols.	Cell culture has the potential to reduce animal testing and spare animal lives. However, the possible use of animal serum raises concerns for animal welfare and human biosafety.	Similar to 2D platforms.
Gene expression	Reflective of animals in vivo. Can differ from humans	Lower expression levels and numerous variations from in vivo and 3D gene expression.	More reflective of in vivo gene expression, thus contributing to better tissue-specific functions than 2D systems.
Morphology	Reflective of animals in vivo. Can differ from humans	Restrictions encountered in 2D environments cause changes in cell morphology and induce an artificial apical–basal polarity.	A three-plane environment allows for the development of complex morphologies.
In vivo imitation	N/A	2D systems do not accurately mimic the natural 3D microenvironment of cells. This leads to misleading and unpredictable data for in vivo responses.	3D cell cultures allow for a better representation of the in vivo organization than 2D systems, resulting in more physiologically relevant data.
Transferability to humans	Controversial. Some very important discoveries for humans had been made using animals. However, animal models are sometimes inefficient to predict human in vivo responses, especially for toxicity studies.	Using human cells minimizes the questionability of transferability of in vitro data to humans and opens doors for personalized medicine.	3D platforms can produce results using human cells in physiological contexts which can lead to high translational potential of the discoveries.
Complexity of environment	Whole organisms are highly complex, thus implying potential unknown interactions.	Low complexity. Often more easily interpretable results.	Intermediate, leading to more relevant data than 2D systems, while controlling most interactions.
Tumour modeling	Helpful models to study tumours.	Cannot accurately replicate characteristics of the tumoural microenvironment.	Suitable for the development of tumour models.
Reproducibility	Not satisfactory, especially in preclinical research.	High reproducibility potential, but decreased by the use of animal serum.	Various 3D techniques offer lower reproducibility than 2D platforms, although high reproducibility is achievable (e.g., hanging drop technique). Reproducibility seems diminished by the use of animal-derived scaffolds or serums.
High throughput agreement	Small animals can be suitable for high throughput screening.	Easily suitable for high throughput screening.	For a long time, difficult to adapt for high throughput screening. New technologies (e.g., tissue chips/microphysiological systems) render them more accessible. Automation is possible but at higher costs.
Vascularization	Reflective of in vivo. Advantageous for tumour and angiogenesis studies.	Lack of vascularization.	Endothelialization of 3D tissues is possible for certain techniques and could improve graft take in addition to being useful for tumour and angiogenesis studies.
Immune system interactions	Presence of interactions. However, immunodeficient models cannot adequately reflect interactions normally encountered with an entirely intact immune component.	Usually no interactions. Low complexity interactions can be encountered in 2D co-cultures with immune cells.	Potential for higher interactions than 2D systems. Incorporation of immunogenic components like immune cells and lymphatic capillaries are being explored to establish 3D systems with more complex in vivo-like interactions with the immune system.

**Table 2 bioengineering-07-00115-t002:** Advantages and disadvantages of 3D cell culture protocols.

Methods		Advantages	Disadvantages	References
Synthetic scaffolds	Metals	Biocompatible Great mechanical properties	Potential poor biodegradability Oxidation and aggregation issues May require to be combined with a polymer Secondary release of metal ions may cause toxicity	[101]
	Ceramics	Osteoconductive and osteoinductive properties (bioactive ceramics) Composition can be similar to the human bone mineral content	Significant brittleness May display inappropriate degradation/resorption rates	[102]
	Polymers	Good tunability of physical properties Low immune response Low production cost High reproducibility Defined purity and composition	Often hydrophobic Lack of cellular recognition patterns for some of them Poor biocompatibility Risk of biodegradation side effects (inflammation, toxicity, etc.)	[101,102]
Natural scaffolds	Polysaccharides	High biocompatible Low toxicity Biodegradable Often contain biofunctional molecules, cell recognition patterns, and adhesion sites on their surface Similarity with native ECM	Limited physical properties Might contain pathological impurities such as endotoxin Difficult to process Properties dependent on extraction and processing procedures	[101,102]
	Proteins	Biocompatible Biodegradable Similarity with native ECM Good interactions with the cells	Limited physical properties Low stability Possible transfer of pathogens Composition varies between batches Unidentified growth factors and bioactive components	[102,103]
	ECM-derived	Natural ECM components Good interactions with the cells Contain biofunctional molecules Minimal processing	Uncertain composition In-between lots variations Risk of unwanted interactions or interferences in signaling pathways	[104]
	Acellular matrix	Preservation of the native ECM High biocompatibility	Incomplete decellularization may generate unwanted immune and inflammatory responses	[105]
Hydrogels	Natural/ Synthetic	High water content Can be made from a large variety of natural or synthetic materials Highly biocompatible Controlled degradation rate Highly tunable Inexpensive Co-culture possible High reproducibility	Limited physical properties Gel-to-gel variations Structural changes over time	[105,106]
Scaffold-free	Spheroids	Simple protocols Highly reproducible Possibility to use multiple cell types Size is easily tunable Inexpensive	Simplified architecture Limited flexibility Limited size Lack of porosity Lack of matrix interaction	[106,107]
	Organoids	Cells can be extracted from a patient Great biomimicking abilities	Has a certain variability Hard to reach in vivo maturity Lack vascularization Can lack key cells types	[107]
	Self-assembly	Cells produce their own ECM Cells can be extracted from a patient Enable the formation of tissue-mimicking organ-specific tissue depending on the cell types used	Requires long culture time Limited size Limited mechanical properties Lack vascularization Incomplete cell differentiation alters biomimicking properties.	[108]

**Table 3 bioengineering-07-00115-t003:** Advantages and disadvantages of basic self-assembly techniques compared with common methods in tissue engineering.

Advantages	Disadvantages
No need of scaffold: no immune or other reaction associated with the graft	Expensive
Good mechanical strength of the reconstruct tissue	The technique is more complicated
Reconstruct tissue present histologically similar to the native tissue	Longer than other techniques

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
