# Peer review of "Innovative Human Three-Dimensional Tissue-Engineered Models as an Alternative to Animal Testing"

_bioengineering, 2020, doi:10.3390/bioengineering7030115_

Round 1
Reviewer 1 Report
In the review "Innovative human three-dimensional tissue-engineered models as an alternative to animal testing" Bédard et al. provide an excellent overview of current possibilities to reduce animal numbers in research and discuss different methods critically. In my mind this review is of high interest to the readers of Bioengineering since pursuing the 3R concept is currently and in future of high importance.
Before I can recommend publication, I have some minor points that should carefully checked by the authors.
- 1.2. Why animal testing: In my mind some details are missing. The authors should e.g. explain that for some research studies the whole organism is necessary / e.g. also discussing the necessity of large animals for preclinical testing / further should be mentioned that in some countries animal experiments have to be performed by law e.g. for approval of some drugs/toxicity studies before performing clinical studies
- 1.4.2. This paragraph is written very critical, I would recommend to also underline that there are already approved in vitro culture methods that can replace animal experiments (e.g. alternatives to the common draize test , at least experiments could be reduced, perhaps in future totally replaced)
- 1.4.2 comes before 1.4.1. - numbering should be adapted
- 1.4.3 refinement - In my mind the reader should get at least a short impression what is meant by this term rather than just providing references.
- 1.6.1 Although the problems about using FCS are explained later in the manuscript, I would recommend to add this issue somewhere shortly in this paragraph.
- I would recommend to put 1.6.4 before 1.6.3
- l. 463 f: "The particular architecture of the scaffold also makes it harder to do cell imaging...." This should be better explained
Author Response
We would like to thank the editor and both reviewers for their time and attention, in order to improve the quality of our manuscript. We highlighted in yellow the modification from the previous version of the manuscript.
Reviewer 1:
In the review "Innovative human three-dimensional tissue-engineered models as an alternative to animal testing" Bédard et al. provide an excellent overview of current possibilities to reduce animal numbers in research and discuss different methods critically. In my mind this review is of high interest to the readers of Bioengineering since pursuing the 3R concept is currently and in future of high importance.
Before I can recommend publication, I have some minor points that should carefully checked by the authors.
- 1.2. Why animal testing: In my mind some details are missing. The authors should e.g. explain that for some research studies the whole organism is necessary / e.g. also discussing the necessity of large animals for preclinical testing / further should be mentioned that in some countries animal experiments have to be performed by law e.g. for approval of some drugs/toxicity studies before performing clinical studies
We agree with the comment and added this to section 1.2:
Moreover, some pathologies or biological phenomenon, requiring interaction between several organs, do require the use of animal as an experimental “unit”: for example, studying metastases. Also, despite intense work of lobbying from opponents to animal testing, many regulatory agencies still require the use of animals in the preclinical testing phases. For example, U.S. federal laws require that non-human animal research occur to show the safety and efficacy of new treatments before any human research will be conducted (U.S. Food and Drug Administration. Investigational New Drug (IND) Application. U.S. Food and Drug Administration. October 5, 2017. Available at: https://www.fda.gov/drugs/typesapplications/investigational-new-drug-ind-application.).
- 1.4.2. This paragraph is written very critical, I would recommend to also underline that there are already approved in vitro culture methods that can replace animal experiments (e.g. alternatives to the common draize test , at least experiments could be reduced, perhaps in future totally replaced)
We added:
Currently, several in vitro protocols, including tissue-engineering derived ones, have been designed to reduce animal use, especially for toxicological testing. For example, protocols using 3D reconstructed human cornea-like epithelium and epidermis models were set up as an alternative to in vivo Draize rabbit eye and skin irritation tests [35]. Several alternatives will be described in section 1.5.
Additional ref 35: Lee M.,Hwang J.H., Lim K.M.; Alternatives to In Vivo Draize Rabbit Eye and Skin Irritation Tests with a Focus on 3D Reconstructed Human Cornea-Like Epithelium and Epidermis Models. Toxicol Res. 2017 Jul;33(3):191-203.
- 1.4.2 comes before 1.4.1. - numbering should be adapted
OK, we did it.
- 1.4.3 refinement - In my mind the reader should get at least a short impression what is meant by this term rather than just providing references.
We agree and to clarify, we added:
Providing a better quality of life to animals is especially important because animals’ discomfort and distress during the experiments can lead to fluctuations in the results and require to repeat the experiences, then increase the number of animal. Briefly, the objective is, not exhaustively, to optimize the methodology applied to animals: this involves reducing, eliminating or relieving their pain or distress, and thereby improving their well-being: to improve transport, breeding and housing conditions, to plan the protocol to avoid stress, to use painless or non-invasive procedures, to provide adequate care before, during and after the operation, to balance anesthesia procedures with regard to pain relief, to minimize the duration of certain studies to the strictly required period, to choose the more appropriate euthanasia procedures. The analyses should be carefully planned to obtain the maximum of data from the experiments.
- 1.6.1 Although the problems about using FCS are explained later in the manuscript, I would recommend to add this issue somewhere shortly in this paragraph.
We agree and added:
Use of animal serum: Even if it is not specific of 2D cell cultures (it is also found in several 3D cell culture models), use of animal serum, especially fetal serum, can induce cell reactions, which differ from the in vivo physiology. For example, it can mask several signaling by activating other pathways. Moreover, animal serum contains not only cytokines and hormones but also remainder of extracellular matrix, bioactive lipids, etc. depending on how the serum had been prepared: the composition of animal serum can differ from lot to lot and provide inconsistent results. All these aspects will be discussed later in the text.
- I would recommend to put 1.6.4 before 1.6.3
OK, we did it
- 463 f: "The particular architecture of the scaffold also makes it harder to do cell imaging...." This should be better explained
We added:
Cells can have limited accessibility for immunostaining. Also, light scattering, refraction and attenuation occur in 3D composite cell-laden gel.

Reviewer 2 Report
This review article describes 3D tissue-engineered models as alternatives to animal models. This is an interesting topic attracting increasing research interest in recent years. The following points should be addressed before it can be accepted for publication:
General comment: A major issue is the use of inaccurate statements, or sentences lacking scientific rigor. many of such issues can be rectified by rephrasing the sentences.
General comment: The English language also needs to be improved. Grammatical errors have to be corrected; sentence structures can be simplified and made clearer.
General comment: Another room for improvement is the classification of models/materials/methods. I found the categories to often overlap with one another, creating confusion and inaccuracy.
General comment: 3D cell culture is not a new concept. This field is moving forward toward models that can better recapitulate the complexity and tissue interactions in vivo. Organoids and microphysiological systems (organ/tissue-on-a-chip systems) are two examples of such efforts. The authors can consider enhancing their discussions on efforts made to engineer complex tissues (with complex structures and functions) and report the recent progress.
Line 818: a few other types of tissues can be constructed using self-assembly approaches. One common example is cartilage. The authors need to modify the figure and/or figure caption.
Line 785 and others: the word schematic in my opinion is more suitable than schema. Please comment.
Line 665: cell amplification should be replaced with cell expansion or cell proliferation
Line 665 and others: in general, I don’t feel it’s appropriate to classify scaffolds into 3 categories: synthetic, natural, and hydrogels. Hydrogels can be either synthetic, natural, or a mix of both. Polymer nanocomposites again can be synthetic or natural or a composite of both. Such classifications can cause confusion and misunderstanding.
Table 2: it’s not fair to use the general term “poor biodegradability” to describe metals. In fact, Mg-based metal implants suffer from fast degradation and are usually surface modified to increase their corrosion resistance. The advantages of ceramics listed only hold true for bioactive ceramics, in particular calcium phosphates. There are also bioinert ceramics such as zirconia and alumina. They are not osteoinductive, nor are they similar to natural bone mineral in terms of composition. What does “Lack of cellular recognition patterns” mean for synthetic polymer? Electrospun scaffolds can exhibit patterns that favor cell adhesion and differentiation. Also there are abundant synthetic polymers with good (not poor) biocompatibility for tissue engineering.
Table 1: gene expression – please also consider the inherent genetic differences between animals and humans. Transferability to humans – the way animal models and 2D culture are described seems to favor 2D cell culture, while in reality animal models in many cases provide more reliable/useful data than 2D culture; also, 3D and 2D culture should be differentiated here. Throughput – 3D culture can be used for high-throughput testing, especially when compared with animal models. Examples are tissue chips/microphysiological systems.
Line 70: section number should be 1.4.1
Line 525: depolymerize
Line 118: alternatives
Line 230: obtained
Figure 2: mice should be changed to mouse to keep animal model name consistent
Figure 1: a more appropriate title would be advantages and applications of common animal models. Large animal models such as dog, pig, and horse are not included.
Line 50: it is not a scientific statement that animals are very similar to humans. The reason they are used for scientific research is their similarities to humans. However, differences, in many aspects, do exist.
Line 73-74: what is the message for the example given? The authors are expected to mention the findings and how they relate to the statement that animals can feel pain and stress.
Section 1.6.3: comparison was not made between 3D cell culture and animal model but cell culture in general and animal models.
Section 1.6.5: if the comparison is made between 3D models and animal models, many disadvantages should be treated as “disadvantages”, such as the use of animal components, cost, time, etc. They are areas that can be improved on, and are drawbacks that are intrinsically difficult to overcome in animal models.
Line 753-754: not all microfluidic devices use mechanical pumps; not all systems have artificial blood vascular network.
Ling 769: please refer to the figure in the paragraph, not in the section title
Table 3: what do other methods mean? Is self-assembly necessarily more complicated, longer, and more expensive?
Author Response
We would like to thank the editor and both reviewers for their time and attention, in order to improve the quality of our manuscript. We highlighted in yellow the modification from the previous version of the manuscript.
Reviewer 2:
This review article describes 3D tissue-engineered models as alternatives to animal models. This is an interesting topic attracting increasing research interest in recent years. The following points should be addressed before it can be accepted for publication:
General comment: A major issue is the use of inaccurate statements, or sentences lacking scientific rigor. many of such issues can be rectified by rephrasing the sentences. General comment: The English language also needs to be improved. Grammatical errors have to be corrected; sentence structures can be simplified and made clearer.
We apologize for our bad English. We use the software “Grammarly” which correct most of the English mistakes and give us a good score after editing. But we did again the operation to improve the text. We also rewrote some sentences to render them shorter and clearer. This remark is a good point, it is sometimes difficult to think well with several concepts at the same time.
General comment: Another room for improvement is the classification of models/materials/methods. I found the categories to often overlap with one another, creating confusion and inaccuracy.
We adopted such a classification from the more synthetic to the more natural and physiologically relevant models. First, it is relatively common in review articles or book chapters published to date but it also fits with our philosophy of the reconstruction of tissues since 35 years. We start with synthetic materials, moving to more natural and finished by using the production of the scaffold by the cells themselves. We did not really understand how classification between synthetic and natural scaffold could be confusing and inaccurate? We understand several construct mix both types of biomaterials but we think that, on the contrary, creating section of functionalization of each type of biomaterials with other types could be source of confusion. Moreover, the fact that biomaterials could be functionalized was specified in the text. We can probably enhance the discussion about this fact and we have changed the text in regard to these concerns.
General comment: 3D cell culture is not a new concept. This field is moving forward toward models that can better recapitulate the complexity and tissue interactions in vivo. Organoids and microphysiological systems (organ/tissue-on-a-chip systems) are two examples of such efforts. The authors can consider enhancing their discussions on efforts made to engineer complex tissues (with complex structures and functions) and report the recent progress.
We agree and added:
4.4 Combination of techniques of production and maturation
Recently, combination of several techniques of production or maturation, such as induced pluripotent stem cells, organoids , bioprinting, composite hydrogels, organ-on-chip, microphysiological systems, mechanical stimuli, innervation etc. (e.g.: [1-5]), give us the opportunity to produce a large spectra of complex organs. These organs could also be used for various applications such as potential transplantation [e.g.: 6] or disease modeling [e.g.: 7-9]. For example, after demonstrating the functionality of hepatic cells bioprinted on collagen gels to drug screening [10], mini-liver was produced using iPSC which differentiated into hepatocyte and self-organize into acini [11]. The bases of the production of liver substitute for transplantation has been laid]: such a tissue required not only right differentiation and organization of hepatocyte but also irrigation by a vascular network and potential reconnection to the host [6]. Another example: gut-on-chip platform has been establish to study various physiological aspect of this complex organ which is the gastrointestinal tract [12].
Additional ref:
- Salerno A, Cesarelli G, Pedram P, Netti PA. Modular Strategies to Build Cell-Free and Cell-Laden Scaffolds towards Bioengineered Tissues and Organs. J Clin Med. 2019 Nov 1;8(11):1816
- Bai J, Wang C. Organoids and Microphysiological Systems: New Tools for Ophthalmic Drug Discovery. Front Pharmacol. 2020 Apr 3;11:407.
- Zhao Z., Vizetto-Duarte C., Moay Z.K., Setyawati M.I., Rakshit M., Kathawala M.H., Ng K.W.Composite Hydrogels in Three-Dimensional in vitro Models. Front Bioeng Biotechnol. 2020 Jun 16;8:611.
- Kaarj K., Yoon J.Y. Methods of Delivering Mechanical Stimuli to Organ-on-a-Chip. Micromachines (Basel). 2019 Oct 14;10(10):700.
- Das S, Gordián-Vélez W.J., Ledebur H.C., Mourkioti F, Rompolas P, Chen I.H., Serruya M.D, Cullen K.D. Innervation: the missing link for biofabricated tissues and organs. NPJ Regen Med. 2020 Jun 5;5:11.
- Kryou C., Leva V., Chatzipetrou M., Zergioti I. Bioprinting for Liver Transplantation. Bioengineering (Basel). 2019 Oct 10;6(4):95.
- Tang H, Abouleila Y, Si L, Ortega-Prieto A.M., Mummery C.L., Ingber D.E., Mashaghi A. Human Organs-on-Chips for Virology. Trends Microbiol. 2020 Jul 13;S0966-842X(20)30184-0
- Santoso J.W., McCain M.L.Neuromuscular disease modeling on a chip. Dis Model Mech. 2020 Jul 7;13(7):dmm044867.
- Sahu S, Sharan S.K.Translating Embryogenesis to Generate Organoids: Novel Approaches to Personalized Medicine. iScience. 2020 Sep 25;23(9):101485.
- Leva, V.; Chatzipetrou, M.; Alexopoulos, L.; Tzeranis, D.S.; Zergioti, I. Direct Laser Printing of Liver Cells on Porous Collagen Scaffolds. JLMN J. Laser Micro Nanoeng. 2018, 13, 234–237.
- Faulkner-Jones, A.; Fyfe, C.; Cornelissen, D.J.; Gardner, J.; King, J.; Courtney, A.; Shu, W. Bioprinting of human pluripotent stem cells and their directed differentiation into hepatocyte-like cells for the generation of mini-livers in 3D. Biofabrication 2015, 7, 044102.
- Steinway S.N., Saleh J, Koo B.K., Delacour D, Kim D.H. Human Microphysiological Models of Intestinal Tissue and Gut Microbiome. Front Bioeng Biotechnol. 2020 Jul 31;8:725.
Line 818: a few other types of tissues can be constructed using self-assembly approaches. One common example is cartilage. The authors need to modify the figure and/or figure caption.
As it is mentioned, this section is related to the tissues which have been obtained using the self-assembly method developed at LOEX by Dr. François A. Auger and coworkers. We recognize, it is a limitation. The title of the figure was changed.
Line 785 and others: the word schematic in my opinion is more suitable than schema. Please comment.
We replaced schema by schematic description
Line 665: cell amplification should be replaced with cell expansion or cell proliferation
We did not find text in line 665, also we presume it is the Fig.5? We changed amplification for expansion.
Line 665 and others: in general, I don’t feel it’s appropriate to classify scaffolds into 3 categories: synthetic, natural, and hydrogels. Hydrogels can be either synthetic, natural, or a mix of both. Polymer nanocomposites again can be synthetic or natural or a composite of both. Such classifications can cause confusion and misunderstanding.
We totally agree with the reviewer. As it was specified by the reviewer, hydrogels can be either synthetic, natural, or a mix of both. This is why we have decided to separate hydrogels from synthetic and natural scaffolds. Also, the properties of hydrogels are different from other scaffolds and many reviews and special issues of scientific journals are dedicated to hydrogels. We moved hydrogels into a new section 2.1.3 after the natural and synthetic scaffolds. We also changed a little the text. We did not want to move the text concerning hydrogels in the sections relating to synthetics or natural scaffolds (and then we would have had to create a new section for mixed?).
Table 2: it’s not fair to use the general term “poor biodegradability” to describe metals. In fact, Mg-based metal implants suffer from fast degradation and are usually surface modified to increase their corrosion resistance. The advantages of ceramics listed only hold true for bioactive ceramics, in particular calcium phosphates. There are also bioinert ceramics such as zirconia and alumina. They are not osteoinductive, nor are they similar to natural bone mineral in terms of composition. What does “Lack of cellular recognition patterns” mean for synthetic polymer? Electrospun scaffolds can exhibit patterns that favor cell adhesion and differentiation. Also there are abundant synthetic polymers with good (not poor) biocompatibility for tissue engineering.
Once again, we agree with reviewer. In order to be synthetic, we have made some questionable generalizations. Moreover, our group recently published an article on the possible use of biodegradable Mg metal stents. We corrected the text to add “potential”. We also modified remarks about ceramics in order to introduce the comment of the reviewer. We also modified the lack of cellular recognition patterns. Generally synthetic polymers did not achieve the level of cellular differentiation of the natural materials. Most of article present cells with a level of differentiation which is very poor. Sometimes relying on one marker or inaccurate markers. It is possible that some of the synthetic polymers induce right cell differentiation but we think we have to generalize to avoid to lose the reader in details.
Table 1: gene expression – please also consider the inherent genetic differences between animals and humans. Transferability to humans – the way animal models and 2D culture are described seems to favor 2D cell culture, while in reality animal models in many cases provide more reliable/useful data than 2D culture; also, 3D and 2D culture should be differentiated here. Throughput – 3D culture can be used for high-throughput testing, especially when compared with animal models. Examples are tissue chips/microphysiological systems.
Text was modified to introduce the remarks of the reviewer.
Line 70: section number should be 1.4.1
OK, we did it.
Line 525: depolymerize
OK, we did it.
Line 118: alternatives
OK, we did it.
Line 230: obtained
OK, we did it.
Figure 2: mice should be changed to mouse to keep animal model name consistent
OK, we did it.
Figure 1: a more appropriate title would be advantages and applications of common animal models. Large animal models such as dog, pig, and horse are not included.
OK, we did it.
Line 50: it is not a scientific statement that animals are very similar to humans. The reason they are used for scientific research is their similarities to humans. However, differences, in many aspects, do exist.
This sentence is derived from the website of the faculty of medicine of the Stanford University. Nevertheless we recognized that it more a popular sentence than a scientific one. We replace it by:
First, non-human animals share genetic and physiologic similarity to humans.
Line 73-74: what is the message for the example given? The authors are expected to mention the findings and how they relate to the statement that animals can feel pain and stress.
We displaced the sentence into 1.4.3 and rephrase it to be more precise:
Animals can feel pain and mental distress. For example, in 2010, a research team correlated the level of pain that mice were experiencing with their facial expressions in order to provide quantifiable data [21].
Section 1.6.3: comparison was not made between 3D cell culture and animal model but cell culture in general and animal models.
We removed “3D” from the title
Section 1.6.5: if the comparison is made between 3D models and animal models, many disadvantages should be treated as “disadvantages”, such as the use of animal components, cost, time, etc. They are areas that can be improved on, and are drawbacks that are intrinsically difficult to overcome in animal models.
The section is not really about comparison between 3D and animal models but the disadvantages of 3D models in the context of pre-clinical studies. The title has been changed to: Disadvantages of 3D cell culture systems as pre-clinical models
Line 753-754: not all microfluidic devices use mechanical pumps; not all systems have artificial blood vascular network.
Agree with the reviewer, we remove the contentious part of the sentences.
Ling 769: please refer to the figure in the paragraph, not in the section title
OK, we did it.
Table 3: what do other methods mean? Is self-assembly necessarily more complicated, longer, and more expensive?
We have changed the title for: Advantages and disadvantages of basic self-assembly techniques compared with common methods in tissue engineering
Because the matrix need to be deposited to construct the scaffold, the time required to produce a tissue is longer than the one required by common methods in tissue engineering which generally rely on the use of prefabricated scaffold whatever its origin. Therefore the cost of production of a tissue using the basic self-assembly technique is higher than the cost of other products made using common protocols in tissue engineering. Not only the cost of cell culture medium must be take into account but also the cost of high qualified personal required for difficult steps such as the stacking step, for example. This technique is also poorly automatable, what is a cause of increasing cost.

Round 2
Reviewer 2 Report
The authors have addressed all questions and comments. The paper can be published as it is.